# Degradation of Azo Dye Solutions by a Nanocrystalline Fe-Based Alloy and the Adsorption of Their By-Products by Cork

**DOI:** 10.3390/ma16247612

**Published:** 2023-12-12

**Authors:** Wael Ben Mbarek, Maher Issa, Victoria Salvadó, Lluisa Escoda, Mohamed Khitouni, Joan-Josep Suñol

**Affiliations:** 1Department of Physics, University of Girona, Campus Montilivi s/n, 17003 Girona, Catalonia, Spain; 2Department of Chemistry, Faculty of Science, University of Girona, 17071 Girona, Catalonia, Spain; victoria.salvado@udg.edu; 3Department of Chemistry, College of Science, Qassim University, Buraidah 51452, Saudi Arabia

**Keywords:** azo dye, redox process, adsorption, cork, wastewater

## Abstract

In this study, the efficiency of mechanically alloyed Fe_80_Si_10_B_10_ in degrading basic red 46 azo dye is investigated. Moreover, the influences of different parameters, such as pH and time, on the elimination of the aromatic derivatives obtained as by-products of the fracture of the azo group are also analyzed. After beginning the reduction to the normal conditions of pH (4.6) and temperature, the experimental findings showed a discoloration of 97.87% after 20 min. The structure and morphology of the nanocrystalline Fe_80_Si_10_B_10_ powder were characterized by SEM and XRD before and after use in the degradation process. The XRD patterns of the Fe–Si–B powder after redox reaction suggest that the valent zero Fe of the alloy is the reducing agent. Powdered cork was then used as a biosorbent for the removal of the by-products generated, resulting in increasing removal percentages from pH 7 (26%) to pH 9 (62%) and a contact time of 120 min. The FTIR spectrum of the cork after adsorption shows a shift of the bands, confirming the interaction with the aromatic amines. The present findings show that metallic powders and natural cork perform well together in removing azo dye solutions and their degradation products.

## 1. Introduction

Azo-type dyes are widely used in the textile industry, among others. During the coloring of the fabric, wastewaters containing dye solutions with different functional groups are generated. The presence of these dyes interferes with the natural ecosystem, including the photosynthetic processes carried out by some organisms [1]. Their presence in water and their degradation products can also generate broader environmental and health concerns [2], as certain azo dyes have been shown to be carcinogenic and mutagenic [3,4].

Azo dyes, which are synthetic, are highly resistant to removal [5], although photodegradation, biodegradation and non-biological degradation are commonly used to this end.

Conventional physicochemical separation methods, such as adsorption, coagulation, ion exchange, membrane separation, etc., have been widely used due to their efficiency in removing dyes from textile wastewater [6,7,8]. Among these, adsorption with granulated or powdered activated carbon has become one of the most effective treatments [9]. However, the high cost of the adsorbent limits its industrial application and, consequently, other adsorbent agents are being investigated. One of the avenues being explored is the use of bio-sorbents, which are eco-friendly, biodegradable, cheaper and abundantly available [10]. Cork, a lignocellulosic natural material, is one such bio-sorbent that has been used for the removal of organic pollutants such as phenolic compounds and pharmaceutical and personal care products [11]. The sorption mechanism involves surface participation, π–π interactions, electrostatic interaction, and hydrogen bonding between two interacting surfaces, termed biosorbent/adsorbent and adsorbate [12].

On the other hand, nanotechnology is an emerging field that is expected to provide alternative technologies for removing contaminants from wastewaters efficiently and economically [13]. It has been shown that the unique qualities of nanomaterials, such as their large pores, high reactivity, ease of dispersion, hydrophobic/hydrophilic properties, and high surface area, make them excellent candidates for wastewater treatments [14,15]. Al-based high-entropy alloys [16], mechanically alloyed Mn–Al binary alloys [17], and zero-valent iron [18,19] all exhibit a notable capacity to degrade azo dyes. Moreover, elements such as Fe, Cr, and Mn, which form BCC crystals, have been found to display substantial activity in these processes. Despite having an FCC crystal structure, aluminum can function as a high-entropy component of an alloy, resulting in the formation of the BCC phase due to its high activity [20]. In addition, mechanically alloyed powders show good efficiency and fast response rates when used as decolorization agents for azo dye aqueous solutions [17,21,22]. Ben Mbarek et al. investigated in this regard how the addition of Fe and Co influences the mechanically alloyed Mn–Al powders’ capacity to degrade azo dyes. The Mn–Al–Fe powders have shown remarkable decolorization efficiency, and their kinetics have been found to be faster than those of Mn–Al-based alloys containing 10% Fe and 10% Co [17].

Among the different chemical methods used, the discoloration of azo dye solutions highlights the redox behavior of metals and alloys with the –N=N– group. Studies to date related to the use of different metals and alloys for this discoloration have investigated structural differences, chemical composition, the morphology and even the production techniques of different metals and alloys [23,24]. Ghasemabadi et al. found that, in the metal powder of the Mn–Al alloy obtained by different production routes and with different structures, the reduction capacity of Mn was favored by the presence of Al, obtaining a high efficiency in the discoloration reaction at both acidic and basic pH [25]. Independently of the influence of these factors and their efficiency, most of these redox processes generate, either totally or partially, aromatic derivatives as by-products. Sulfonated and non-sulfonated aromatic amines can become more toxic to organisms than the azo dye from which they are derived.

The objective of the present study is, on the one hand, to analyze the efficiency of the reduction of the azo group of the dye basic red 46 (BR46) by the nano-crystalline metal alloy Fe80Si10B10 (at. %), and, on the other hand, to determine the efficacy of granulated cork as a bio-adsorbent of aromatic amines derived from the fracture of –N=N–. For the reduction reaction, the rate and efficiency of the reducing agent are studied under initial pH conditions corresponding to the hydrolysis of the dye (pH = 4.6) and in the absence of carbonate mediators, whereas the influence of pH and time are studied in the case of the adsorption process. Hence, the developed method is based on an efficient, fast, environmentally friendly and sustainable system for the discoloration of aqueous solutions of basic red 46 (BR46) azo dye.

## 2. Materials and Methods

### 2.1. Materials and Characterisation Techniques

To achieve the required composition (Fe_80_Si_10_B_10_ (at. %)), pure elemental powders of Fe (99.9 wt. %), Si (99.9 wt. %), and B (99.9 wt. %) were separately weighed and mixed. At room temperature and in an argon atmosphere, mechanical alloying for 50 h was carried out in a high-energy planetary ball mill (Type P7). Ball milling experiments were carried out in a hardened steel container. The milling speed was set at 500 rpm, and the ball-to-powder weight ratio was kept at 0.47. Powder agglomeration and sticking to the container walls and balls were prevented by using a milling sequence that involved 10 min of milling followed by 5 min of idle time.

Granulated cork, supplied by the Cork Centre (Palafrugell, Spain), was sifted to separate out powder particles of <2 mm. The cleaning procedure consisted of putting an amount of cork into contact with 10 mL of ultrapure water in 25 mL glass tapered tubes, which were then placed in a horizontal rotatory mixer (DINKO, Barcelona, Spain) at 20 rpm for 30 min. After cleaning and air-drying the cork, it was put into contact with the test solutions. Basic red 46 (C_18_H_21_Br_N_6) was purchased from Biosynth AG (Staad, Switzerland).

The structure of the Fe–Si–B powders, before and after azo fracture, was analyzed by X-ray diffraction (Siemens D-500 equipment with Cu-Kα radiation, Forchheim, Germany) with the aim of observing the microstructural changes originating in the alloy and identifying the phases. The morphologies of the mechanically alloyed powders before and after the reduction process were examined by scanning electron microscopy (SEM) using a DSM960A ZEISS microscope (Zeiss, Oberkochen, Germany) in secondary electron mode, operating at a voltage of 15 kV. Fe concentrations in solution after the redox process were analyzed by inductively coupled plasma optical emission spectroscopy (ICP-OES) (Agilent 1500, Agilent, Santa Clara, CA, USA).

### 2.2. Experimental Procedure

The reduction experiments were conducted in sets of 11 samples of 20 mL of basic red 46 aqueous solution with initial concentrations of 200 mg·L^−1^ by adding 0.1 g of Fe-based alloy to the dye solution. The natural pH of the solutions was 4.6. The mixtures were continuously agitated, and samples were taken at different times. The UV–vis spectrum of each sample was recorded after filtration by measuring the absorbance between 200 and 800 nm with a UV–vis spectrophotometer to evaluate the efficiency of the degradation process over time. A calibration curve was constructed by measuring the absorbance of standard solutions of RB46, ranging from 1 to 210 mg·L^−1^. The pH of the samples was measured using a pH-meter Basic 20^+^ (Crison Instruments, Alella, S.A., Spain). The filtered solution was also analyzed via ICP-OES to determine the presence of dissolved ions originating from the Fe-based alloy.

The process of the azo reduction was monitored by following the changes in the absorbance bands at 400–800 nm corresponding to the n→ π^∗^ transition of the azo group of BR46 dye [26]. The absorbance at 200–400 nm is attributed to the n→π^∗^ transition of the benzene ring, which represents the aromatic part of the azo dye molecule, and the decrease in its intensity is due to the degradation process that takes place on the surface of the metal alloy resulting in the complete decoloration of the solution.

The removal of the by-products generated by the reduction of basic red 46 (BR46) was investigated by the adsorption process. The sorption experiments were performed in triplicate in 25 mL tubes by adding 0.03 g of cork to 15 mL of the solution resulting from the reduction process, which was previously filtrated and diluted to 50%. The mixture was then shaken using a rotational agitator for a maximum of 2 h. The samples were then withdrawn at prefixed times and centrifuged to separate the adsorbent from the solution. The absorbance of the resulting solution was measured at 248 nm to monitor the changes in the concentration of the residual aromatic amines from the supernatant. The cork removal efficiencies were calculated from the absorbances at t = 0 of the adsorption process and those measured from the samples collected at the prefixed times. The changes in the cork surface due to the adsorption of the aromatic amines that formed as a result of the chemical reduction of the azo dye were investigated by recording the FTIR spectra of cork before and after adsorption. The effect of the pH on the sorption process was studied at pH values of 5, 7, 9, and 10. The pH values of the test solutions before adding the cork were adjusted by adding either HCl (0.1M) or NaOH (0.1M). All experiments were performed in triplicate at room temperature.

## 3. Results and Discussion

XRD diffraction patterns at room temperature were performed to determine the microstructure of the metallic powders, produced by mechanical alloying, using Cu–K_α_ radiation before and after the reduction process. Figure 1a presents the XRD patterns of the Fe–Si–B powders before the reduction process. We can see there is one crystalline phase, the α-Fe solution (bcc Fe rich phase with Si and B in solid solution). On the other hand, several techniques have been developed in addition to the Williamson–Hall [27] and Halder–Wagner [28] methods to separate the microstructural characteristics. As is known, line broadening (*β*) is the result of lattice distortions and crystallite size [29]. Thus, the total line broadening *β* can be expressed as:(1)$$\beta =\frac{k\lambda }{D}\frac{1}{\cos\theta }+4\varepsilon \tan\theta$$
where ε is the lattice strain and *D* represents the size of the crystallite. The size contribution is the first component and the lattice distortion is the second in the right-hand part of Equation (1). The Scherrer equation states that *β* = *kλ/(<D>cosθ*), *k* is close to 1 if the peak broadening is mainly caused by a finite crystallite size [29]. In the current investigation, the average crystallite size of the mechanically alloyed α-Fe–Si–B sample was calculated using the plot model of Equation (1). The diffraction peak broadening of (110), (200), (211), (220), and (310) peaks was used as a basis for computing crystallite size and lattice strain. In fact, Equation (1) is a linear equation that accounts for the isotropic nature of crystals. The inverse of average crystallite size <*D*> can be derived from the intercept, and the lattice strain (ε) can be derived from the slope straight line of the plot made with (4.sinθ) along the *x*-axis and (*β.cos θ*) along the *y*-axis. The calculated average crystallite size was found to be approximately 12 ± 2 nm with a lattice strain of 0.35%. The arrangement of atoms within the crystal lattice is primarily responsible for the lattice distortion in the crystallites, which is the source of the lattice strain. However, size refinement and internal–external stresses that cause lattice strain also led to the creation of numerous structural defects (point defects such as vacancies, stacking faults, grain boundaries, dislocations, etc.) in the lattice structure [30,31].

Figure 1b shows the XRD patterns of the Fe–Si–B powder after the redox reaction. The second phase that can be seen in the spectrum has been identified as a hydrated iron oxide. The formation of FeOOH suggests that the active reducing agent is zero-valent iron. It is possible that the iron oxidation process is activated due to small changes in the electrical potential caused by different structural or morphological environments of the iron. The oxidation process of iron in aqueous medium generates Fe (II) and hydrogen, which remains adsorbed on the surface of the alloy. Fe (II) is slowly oxidized to Fe (III), whereas hydrogen intervenes by fracturing the azo group on the surface of the metal. Under standard conditions, water acts as an oxidant, favoring the oxidation of Fe and the formation of hydrogen. The oxidizing character of water increases in acidic conditions, producing more hydrogen and, consequently, increasing the decolorization kinetics [32].

The SEM micrographs of the Fe80Si10B10 powders before and after the reduction process of the azo group, taken at a magnification of 90 μm and 10 μm, respectively, are shown in Figure 2. Figure 2a illustrates the particle size distribution and surface morphology of the powders before the reduction process. The metallic alloy powder shows an essentially spherical morphology with different diameters ranging from 5 to 40 µm, which are typical of ball-milling powders [33]. The particles are distributed relatively uniformly and there are aggregates where fine particles are stuck to the surfaces of coarse ones. Figure 2b shows particles of different sizes and more irregular surfaces and shapes than before the reduction process.

The surface morphology (at a higher magnification) of the ball-milled Fe80Si10B10 powder before reduction is shown in Figure 2c,d. It can be seen that there is some aggregation present in addition to a generally uniform distribution of particle sizes, and their surfaces are heavily corrugated. Figure 2d depicts the surface shape of the Fe–Si–B powder after interaction with basic red 46 solutions. The feasibility of use in the formation of micro- or nanobatteries is validated by the morphological investigation. On all surfaces of the Fe_80_Si_10_B_10_ powder particles, the reaction products are uniformly distributed nano-bristles. Numerous corrosion pits can also be seen on the surfaces of powder particles, indicating that the basic red 46 degrading process was responsible for the pitting corrosion on the alloy particles. Similar observations were made by Ben Mbarek et al. in their most recent investigation [32]. Likewise, EDX microanalysis confirmed the presence of oxygen after the redox process.

Moreover, the analysis by ICP-OES of the dye solutions after 20 min of contact with the Fe–Si–B power confirms the presence of Fe at a concentration of 0.030 ± 0.01 mg·L^−1^, showing that Fe is the active agent of the alloy and that the surface of the metal powder suffers modification during the reduction process. The decrease in the concentration of the Fe on the solid surface due to the oxidation of the metal leads to an increase in the concentration of B and Si [32]. The enrichment of metalloid elements is indicative of the formation of a non-homogeneous surface layer, which can balance the effect of precipitates, making the contact of the inner zero-valent iron (ZVI) atoms with dye solutions more effective, and thus allowing the rapid oxidation kinetics of the reducing agent. With regard to the iron oxidation process, the mechanism may involve the formation of nano- or micro-batteries between iron and superficial silicon oxide.

In Figure 3a, we can observe the evolution of the redox process in the registered absorption spectra (200–800 nm) over time. Almost complete discoloration of the solution was achieved at pH = 4.6 and room temperature 20 min after the start of the reaction. The characteristic peak associated with the –N=N– dye corresponds to λ = 534 nm. The changes observed in the UV–vis spectra indicate that the concentration of azo dye (200 mg·L^−1^) at t = 0 min began to slowly decrease after the first 5 min. However, the decrease in the concentration of azo dye was particularly accentuated after 15 min, reaching a minimum of 4.3 ± 0.3 mg·L^−1^ (*n* = 3) at 20 min. The rapid decolorization makes Fe–Si–B one of the most efficient reductors for decolorization at room temperature. The nanocrystalline Fe–Si–B powder shows a redox activity that is comparable to that of amorphous ZVI powders [25].

Along with the decrease in the dye concentration in the solution, an absorbance peak appears at ∼248 nm on the UV–vis spectrum as a result of the azo dye degradation, which corresponds to the aromatic amine groups [34]. Therefore, as the azo reduction process advances, the intensity of this peak is gradually enhanced. The basicity of the generated by-products resulted in an increase in the pH of the solution during the reduction process, from an initial pH of 4.6 to pH 6.5 after 5 min, and to 7.3 after 15 min. Figure 3b shows the degradation ratio as a function of time for three chromophores of the dye and reaction products at their respective wavelengths of maximum signal. It can be observed that the evolutions of the bands at the different wavelengths differ greatly. The peaks associated with the azo group (λ = 534 nm) and the aromatic rings (λ = 290 nm) of the dye decrease, while the characteristic peak of the aromatic amines (λ = 248 nm) increases. The evolutions of the amine aromatic and –N=N– bands, which stabilized at 30 min after starting the redox process, can also be seen.

In order to study the removal of the degradation by-products using cork, a specific amount of cork (0.03 g) was added to 15 mL of the solution resulting from the reduction process, which had been previously filtrated and diluted to 50%. The absorption spectrum of this solution was recorded before adding the cork and after stirring the mixture for 60 min at room temperature. Figure 4a shows a comparison of the spectral UV changes between the solution with and without the sorbent, under the same conditions. It can be seen that the intensity of the peak of aromatic amines at 248 nm was reduced in the presence of cork, and that all the peaks linked to the dye disappeared from both spectra. Hence, the changes in the concentration of the residual aromatic amines from the supernatant were monitored by measuring the absorbances of the solution at 248 nm.

The aromatic derivatives were partially removed from the resulting solution by an adsorption process. This result confirms that the complete removal of azo dyes from aqueous solutions by Fe requires the reduction or cleavage of the azo bond accompanied by the elimination of aromatic amine by-products [25]. As can be seen in Figure 4a, the adsorption process of aromatic amines by cork reduced these by around 50%. The aromatic rings and carboxyl and hydroxyl groups of suberin and the lignin of cork explain the interactions with organic pollutants. Raw cork has a high capacity to absorb hydrophobic compounds (log K_ow_ > 4), but is less successful in the case of hydrophilic compounds (log K_ow_ < 2) [35]. This behavior is associated with the interaction of the aromatic components of lignin with the aromatic moieties of the adsorbed compounds via π–π interactions, as is the case in phenolic compound sorption [11]. In the present case, the aromatic amine by-products act as π acceptors in forming π+–π electron donor–acceptor interactions with the π electron-rich cork.

The adsorption percentage increases from pH = 5 (6 ± 4%) to pH = 7 (26 ± 3%), reaching its maximum efficiency (62 ± 3%) at pH 9, before slightly decreasing at pH 10 (56 ± 4%). When analyzing the weak basic behavior of aromatic amines in aqueous medium, whose pK_as_ vary significantly, it seems that at this pH the amine group is not protonated, and the lone pair of the nitrogen of this group is involved in resonance forms favoring π–π interactions with cork. Moreover, the progression of the adsorption efficiency of cork over time was studied, resulting in an equilibrium time of 120 min.

Figure 5 presents the FTIR spectra of cork before and after adsorption at pH = 9. The peaks associated with the cork before adsorption could be assigned as follows: 3425 cm^−1^ (γ OH); 2919 and 2848 cm^−1^ (γ C–H) for asymmetric and symmetric C–H and CH_2_ (γ C=O); 1747 cm^−1^ and 1635 cm^−1^ for (δ C–O) 1036 cm^−1^, where γ represents a stretching vibration and δ, a blending vibration. An obvious change was observed on the spectrum of cork after the adsorption of the aromatic amines. The absorption bands at 3425, 2919, 2848, 1747, 1635, and 1036 cm^−1^ before adsorption shifted, respectively, to 3412, 2912, 2842, 1730, 1620, and 1025 cm^−1^ due to the adsorption process. These shifts might be attributed to the π–π interactions of the aromatic amines with the aromatic ring and carboxyl and hydroxyl groups of suberin, and the lignin of cork [11].

Therefore, the results show the viability of using both metallic and cork powdered particles as a combined pathway to degrade dyes in wastewater.

## 4. Conclusions

In this study, cork and nanocrystalline Fe80Si10B10 (at. %) powders have been used to degrade the BR46 azo dye and eliminate aromatic amines from the derived aqueous solutions. The metallic particles provoked azo bond cleavage and decolorization through a reduction process. The redox process induced the formation of a hydrated oxide on the surface of the Fe-based alloy, but this layer failed to completely inhibit the activity of the underlying zero-valent iron. Despite this, it was found that the reduction reaction continued until the dye concentration fell to around 4.3 mg/L at 20 min. However, aromatic amines are undesired by-products of this process that need to be eliminated. Cork has been shown to be an efficient biosorbent that can be used to remove the generated degradation by-products, achieving removal percentages of 62% at pH = 9 and an equilibrium time of 120 min.

The results of our study indicate that the use of natural cork combined with metallic powders provides an efficient means of removing azo dyes and their degradation by-products from industrial effluents.

## Figures and Tables

**Figure 1 materials-16-07612-f001:**
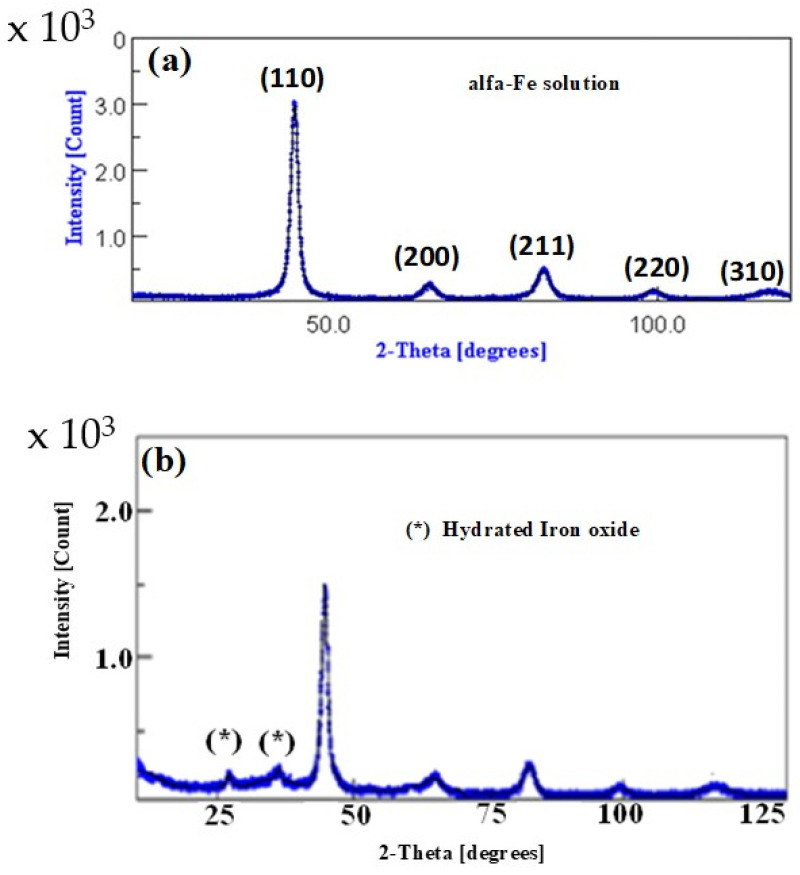
X-ray diffraction patterns of Fe–Si–B (**a**) before the reduction process, and (**b**) after the reduction process.

**Figure 2 materials-16-07612-f002:**
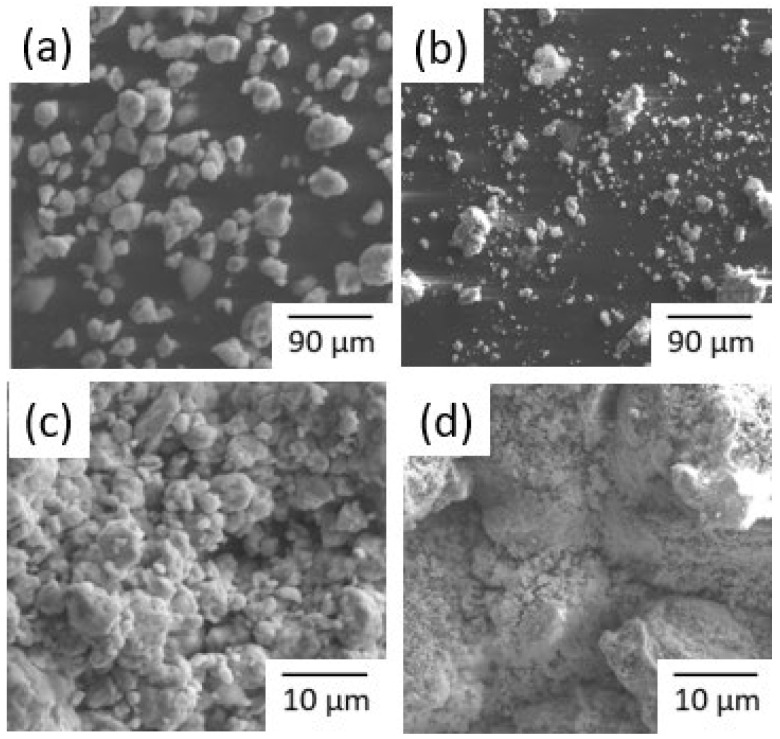
Particle morphologies of the mechanically alloyed Fe–Si–B sample: (**a**,**c**) before and (**b**,**d**) after the decolorization process.

**Figure 3 materials-16-07612-f003:**
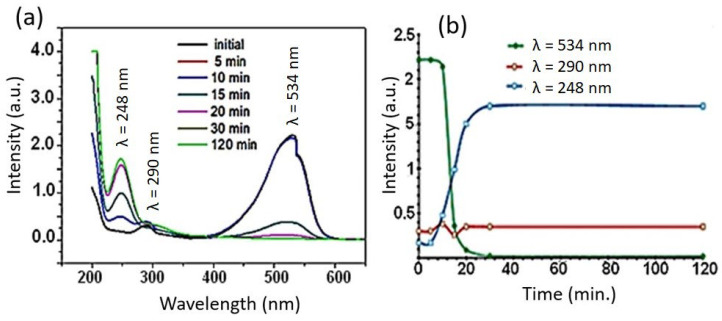
(**a**) The changes in the UV–vis spectra of BR46 azo dye over time (initial, and 5, 10, 15, 20, 30 and 120 min) during the reduction process using Fe–Si–B alloy. (**b**) Changes in the different bands (534 nm, 290 nm and 248 nm) of BR46 spectra during the reduction process with Fe_80_Si_10_B_10_ powders. For clarity, only the spectrum of one of the experiments performed is depicted.

**Figure 4 materials-16-07612-f004:**
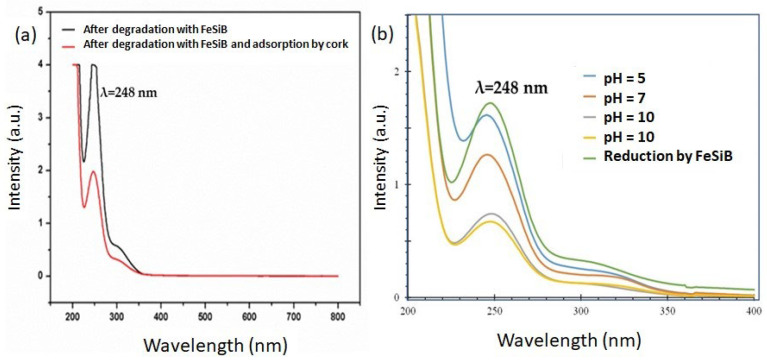
(**a**) Comparison of the UV –vis spectrum of a BR46 solution after its reduction by Fe_80_Si_10_B_10_ with the same solution after adsorption by cork for 120 min and (**b**) the effect of the pH on the removal of aromatic amines using cork. For clarity, only the spectrum of one of the three experiments performed is depicted.

**Figure 5 materials-16-07612-f005:**
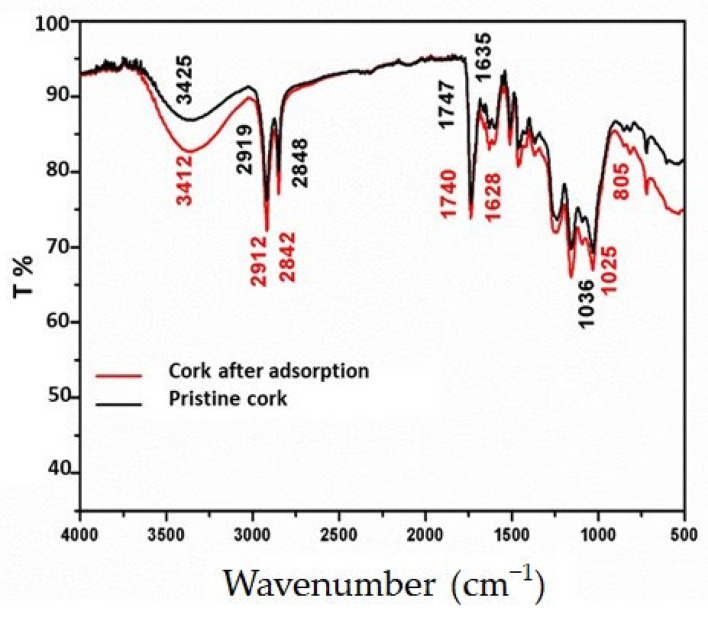
FTIR spectra of pristine cork and cork after adsorption at pH 9 of the aromatic amines resulting from the degradation of BR46 by Fe–Si–B powder.

## Data Availability

The raw data will be made available on reasonable request.

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
