# Peer review of "Degradation of Azo Dye Solutions by a Nanocrystalline Fe-Based Alloy and the Adsorption of Their By-Products by Cork"

_materials, 2023, doi:10.3390/ma16247612_

Round 1

Reviewer 1 Report

Comments and Suggestions for Authors

Article: Degradation of azo dye solutions by combining metallic and cork powders for publication in Materials is good. The subject is interesting. This work presents an efficient, environmentally friendly and sustainable system for the discoloration of solutions of the azo dye Basic Red 46 (BR46). The system is based on the combination of a chemical process, based on the use of nanocrystalline Fe based alloyed powders and a physical process using cork powder as a natural adsorbent agent. The influence of different parameters on the physical process such as pH and time, are controlled to analyze the efficiency of elimination of the aromatic derivatives obtained as by-products of the fracture of the azo group. The experimental results show a discoloration of 97.87% 20 minutes after starting the reduction to the natural conditions of pH (pH = 4.6) and temperature. The work determines that the reducing agent is the valent zero Fe of the alloy and, consequently, the reduction reaction kinetics would improve at more acidic pH. The efficiency of the adsorption of aromatic derivatives, as a consecutive stage to reduction, improves with the increase in pH obtaining a maximum of 48% adsorption at pH = 9 and for a time of 60 minutes with continuous agitation. The absorption efficiency increases if both processes are performed simultaneously with an efficiency of 51.4% at initial pH of 4.6 and at room temperature without appreciable modifications in the kinetics of the redox reaction. This work is a contribution to the development of an economical method that allows recycling natural organic products as adsorbent agents of aromatic derivatives and mixing them with Fe-based alloys as redox agents without pH or temperature modifications due to discoloration of azo dye solutions. The presented article is interesting and a construction of article is logical. The work is relevant and practical. Clarity of expression and communication of ideas, readability and discussion of concepts is good.

However, some corrections are needed:

1.      Figures 2, 4 and 5 should be corrected.

2.      An interesting addition to this publication could be the calculated size of crystallites from XRD results by, for example, Scherrer equation.

3.      I did not find SEM, EDS and FTIR results described in the publication, this should be supplemented.

4.      It would be necessary to unify units according to the Si system.

5.      Novelty elements should be better highlighted in the introduction. Papers should be cited in Introduction section; for example:

Author Response

We agree the referee comments. We send our answers in a world file.

Reviewer 2 Report

Comments and Suggestions for Authors

·         Method to remove dye should be specify in title, specific chemical and physical methods as reported.

·         The abstract is fine, but try to shorten it, if possible. Only include the important findings.

·         The current advanced treatment of dyes should be reviewed further in Introduction section. The current Introduction is very superficial. The problem with advanced treatment can be specify that led to your current works.

·         Each instrumental analysis should be elaborated in terms of procedures.

·         Also, the preparation materials are unclear. What materials that carried out chemical treatment as indicated in abstract?

·         What are the lamda in Fig 2 (b)?

·         Figures are unclear. Fig 2 and 4 especially.

·         Why fig 9 gives the best absorption? How the chemical degradation played role during the adsorption?

·         How do you know that the dye removal was because of adsorption by physical or chemical degradation

·         Please compare the results with other works.

Author Response

(The authors gave the same response as above.)

Reviewer 3 Report

Comments and Suggestions for Authors

There are several unclear issues in this manuscript. There are no statistics. Results are missing. It is not ready for publication. It is hard to believe that six authors allowed this unfinished work to be submitted.

In the Abstract, "The work determines that the reducing agent is the valent zero Fe of the alloy and, consequently, the reduction reaction kinetics would improve at more acidic pH" statement has no backing in the text/experiments.

There are no results for this: "The absorption efficiency increases if both processes are performed simultaneously with an efficiency of 51.4% at initial pH of 4.6 and at room temperature without appreciable modifications in the kinetics of the redox reaction" in the Results and Discussion section. The experiment description in Methodology is, in my opinion, in the wrong place.

No results for morphology are presented, but the SEM use is described in the Methodology. The same goes for FT-IR group designation.

Parts of the Methodology should be moved to Results and Discussion. Paragraph on UV-Vis regions and the following paragraph on the 'simultaneous' procedure since, at the end of R&D, the diagram shows the two paths.

There is a statement that 11 samples were tested, so a statistical analysis should be presented.

 In general, the Results and Discussion section is missing a literature-based discussion. There are very few references and many statements of things happening without clear proof. Please add related literature.

You state that Figure 1B shows "significant differences," but the differences are not noted in the text. The shifts in 2-theta, if they exist, should be indicated either in the text or the figure. As it is, it is hard to tell if there is a change in the peak positions (the scale is different in both sections; Figure 1B seems to have more noise). Also, "significant differences" are typically based on statistical analysis, which is absent here. The discussion in this section is missing literature support – how do you know this happens when you don't analyze the results specifically?

In which table/figure are you showing ICP-OES results?

Figure 2A shows confusing colors; please change the navy blue (120 min) line to a different color to make it stand out and easier to differentiate from the black (initial) line. In the text, please add peak assignments. Somewhere in the Methodology, the general regions were mentioned, but here, a clear indication should be made based on the compound studied.

In Figure 2B: why are you following absorbance of 200nm and not 238nm, which is a clear transition peak?

Figure 4 has low resolution. What about the statistical analysis here? What was the SD of these measurements? Between about 30 to 45% change might or might not be significant…

The discussion following Figure 4 is not wrong; it is just missing a literature connection. However, you didn't present an explanation of why, at pH 10, the absorption drops again (assuming the statistics prove it).

Figure 5 is missing SD and statistics. What is known about the cork absorption characteristics? What did other authors show?

Figure 6 is random. Why is it there? What purpose does it serve? I see that this manuscript showed the diagram's left path, but what is the right path of the diagram?

Comments on the Quality of English Language

Subject-verb disagreement: 

To 15 mL the resulting solution of the reduction process, previously filtrated, and diluted to 50% were mixed with 0.03 g of cork

Not English conjunction:

The decrease in the concentration of Fe on the solid surface due to oxidation of the metal leads to an increase in the concentration of B i Si.

Author Response

(The authors gave the same response as above.)

Round 2

Reviewer 1 Report

Comments and Suggestions for Authors

Accept in present form.

Reviewer 2 Report

Comments and Suggestions for Authors

All comments have been addressed.

Reviewer 3 Report

Comments and Suggestions for Authors

Congratulations, authors, on significantly improving the manuscript. You have addressed the concerns previously mentioned. At this point, the reduction of the dye is well-explained and proven.